# The Influence of the Matrix on the Apatite-Forming Ability of Calcium Containing Polydimethylsiloxane-Based Cements for Endodontics

**DOI:** 10.3390/molecules27185750

**Published:** 2022-09-06

**Authors:** Paola Taddei, Michele Di Foggia, Fausto Zamparini, Carlo Prati, Maria Giovanna Gandolfi

**Affiliations:** 1Biochemistry Unit, Department of Biomedical and Neuromotor Sciences, University of Bologna, Via Irnerio 48, 40126 Bologna, Italy; 2Endodontic Clinical Section, Unit of Odontostomatological Sciences, Department of Biomedical and Neuromotor Sciences, University of Bologna, Via San Vitale 59, 40136 Bologna, Italy; 3Laboratory of Biomaterials and Oral Pathology, Unit of Odontostomatological Sciences, Department of Biomedical and Neuromotor Sciences, University of Bologna, Via San Vitale 59, 40136 Bologna, Italy

**Keywords:** polydimethylsiloxane, crosslinking, apatite, bioactivity, bioglass, GuttaFlow Bioseal, GuttaFlow 2, RoekoSeal, vibrational IR and Raman spectroscopy, endodontic sealer, root filling materials, calcium silicates (CaSi), hydroxiapatite (HA), calcium phosphate dihydrate (DCPD)

## Abstract

This study aimed to characterize the chemical properties and bioactivity of an endodontic sealer (GuttaFlow Bioseal) based on polydimethylsiloxane (PDMS) and containing a calcium bioglass as a doping agent. Commercial PDMS-based cement free from calcium bioglass (GuttaFlow 2 and RoekoSeal) were characterized for comparison as well as GuttaFlow 2 doped with dicalcium phosphate dihydrate, hydroxyapatite, or a tricalcium silicate-based cement. IR and Raman analyses were performed on fresh materials as well as after aging tests in Hank’s Balanced Salt Solution (28 d, 37 °C). Under these conditions, the strengthening of the 970 cm^−1^ Raman band and the appearance of the IR components at 1455–1414, 1015, 868, and 600–559 cm^−1^ revealed the deposition of B-type carbonated apatite. The Raman I_970_/I_638_ and IR A_1010_/A_1258_ ratios (markers of apatite-forming ability) showed that bioactivity decreased along with the series: GuttaFlow Bioseal > GuttaFlow 2 > RoekoSeal. The PDMS matrix played a relevant role in bioactivity; in GuttaFlow 2, the crosslinking degree was favorable for Ca^2+^ adsorption/complexation and the formation of a thin calcium phosphate layer. In the less crosslinked RoekoSeal, such processes did not occur. The doped cements showed bioactivity higher than GuttaFlow 2, suggesting that the particles of the mineralizing agents are spontaneously exposed on the cement surface, although the hydrophobicity of the PDMS matrix slowed down apatite deposition. Relevant properties in the endodontic practice (i.e., setting time, radiopacity, apatite-forming ability) were related to material composition and the crosslinking degree.

## 1. Introduction

Apical periodontitis represents the main disease in endodontic treatments: bacterial proliferation due to untreated dental caries causes an inflammatory lesion of the tooth root. Root canal therapy is a dental procedure used to treat the infected root canal and prevent further microbial contamination [1].

Several endodontic sealers have been designed to perform root canal fillings in obturation procedures [1]. In this context, Grossman [2] listed the properties of an ideal sealer: good adhesion to the canal wall, hermetic seal, radiopacity, ease of mixing, no shrinkage on the setting, no tooth staining, bacteriostatic activity, slow setting, insolubility in host tissue fluids, biocompatibility with the periradicular tissue, and solubility in common solvents, allowing for removal when necessary.

Despite technological advancements of recent years, no sealer has yet satisfied the whole set of Grossman’s criteria. Several sealers have been developed, which may be classified into different groups according to their chemical composition and structure: zinc oxide–eugenol-based, resin-based, glass ionomer-based, silicone-based, calcium hydroxide-based, and bioactive endodontic sealers.

Zinc oxide–eugenol-based sealers (such as Pulp Canal Sealer and Argoseal) were some of the most clinically used materials with cold obturation techniques. These sealers demonstrated some drawbacks, including cytotoxicity when extruded over the apex [3] and inhibition of the polymerization phases of adjacent methacrylate-based materials [4]. Resin-based sealers are divided into methacrylate-based or epoxy resin-based. Methacrylate-based sealers (such as Endorez) are no longer used as they have demonstrated critical in vivo degradation, with loss of the endodontic seal and long-term failures [5]. Differently, epoxy resin-based sealers (such as AH Plus) are widely used in warm obturation techniques. As drawbacks, these sealers are highly hydrophobic and require the absence of moisture in the root canal [3]. Glass ionomer-based sealers (such as Ketac Endo) demonstrated biocompatibility and the ability to bond to dentin, but they are no longer used due to detrimental leakage and disintegration [3]. Calcium hydroxide-based sealers (such as Seal Apex or Apexit) revealed high biocompatibility, but are seldomly used nowadays due to their solubility [6].

Among the materials developed as endodontic sealers, polydimethylsiloxane (PDMS) is a biocompatible polymer extensively used in biomedicine [7,8,9], thanks to its valuable characteristics, i.e., chemical inertness, thermal stability, oxidation resistance, and good mechanical properties [8].

The class of bioactive endodontic sealers typically includes composite materials in which a polymeric matrix is added with a bioactive bioceramic, such as calcium silicates, calcium phosphates (i.e., hydroxyapatite or dicalcium phosphate dihydrate), and bioglasses [9,10,11,12]. In particular, hydroxyapatite (HA) exhibits a chemical composition and crystalline structure similar to living bones and high osteoconductivity in biological environments [9,13,14]. Dicalcium phosphate dihydrate (DCPD) was added to calcium silicate cement and bioresorbable polymers to increase their biological and apatite-forming performances [15]. BioRoot RCS, a tricalcium silicate-based endodontic cement, was reported to induce hard tissue deposition, in vitro production of angiogenic and osteogenic growth factors, and antimicrobial activity [16]. The use of SiO_2_-CaO-Na_2_O-P_2_O_5_ bioglasses (such as 45S5 or 55S4) has been introduced for their excellent bioactivities and promotion of new bone formation in vivo [17]. Bioactive ceramics can undergo interactions with the surrounding tissue and are capable of forming hydroxyapatite or carbonated apatites on their surfaces [18]. Moreover, when a bioactive sealer is compacted against dentin, a dentin-sealer interface layer forms in the presence of calcium and phosphate ions. Therefore, studies evaluating the bioactivity of different types of sealers are essential since bioactivity is considered crucial for relevant functional properties, such as sealing ability, osteoconductivity, and biocompatibility [19].

In this context, the present study aims to gain more insights into the in vitro bioactivity of GuttaFlow Bioseal, a bioglass-containing PDMS-based root canal sealer, in comparison with similar PDMS-based commercial cement not containing calcium silicate and/or other bioactive components, i.e., RoekoSeal and GuttaFlow 2. This property has already been described in the literature [10,11,12], although the role of the matrix in mineralization processes has not been clarified yet. In this study, vibrational IR and Raman spectroscopy were used to characterize the materials more deeply at a molecular level and to monitor the apatite-forming abilities of the cement and the structural modifications occurring at the interface upon immersion in Hank’s Balanced Salt Solution (HBSS), chosen as simulated body fluid. In an attempt to improve bioactivity, GuttaFlow 2 was added with DCPD, HA, or BioRoot RCS.

## 2. Results

### 2.1. Fresh Samples

#### 2.1.1. Commercial Samples

Figure 1 shows the average IR spectra of orange (A) and white (B) pastes of RoekoSeal, GuttaFlow 2, and GuttaFlow Bioseal. Although clinicians use the sealers as the mix of the two pastes, we decided to analyze them separately to gain information on the relative amounts of the reactive sites present in the different cements and, thus, on the crosslinking degree.

All the spectra were dominated by the bands of PDMS at about 2963 cm^−1^ (CH_3_ antisymmetric stretching), 2905 cm^−1^ (CH_3_ symmetric stretching), 1450 cm^−1^ (CH_3_ antisymmetric bending), 1411 cm^−1^ (CH_3_ symmetric bending), 1258 cm^−1^ (CH_3_ symmetric deformation), 1078 cm^−1^ (antisymmetric Si-O-Si stretching), 1010 cm^−1^ (symmetric Si-O-Si stretching), 866 cm^−1^ (CH_3_ symmetric rocking), 790 cm^−1^ (CH_3_ antisymmetric rocking, Si-C antisymmetric stretching), 740 cm^−1^ (Si-C symmetric stretching, Si-C antisymmetric bending, and Si-O bending) [20,21,22,23]. The latter band is also assignable to monoclinic zirconia together with spectral features below 600 cm^−1^ [24]. Band wavenumbers and assignments are summarized in Appendix A.

The spectra of the orange pastes of all the sealers show as distinctive features the bands at about 2160 and 910 cm^−1^, assignable to Si-H stretching and bending, respectively [21,23]. These bands were found to decrease in intensity along with the series: GuttaFlow Bioseal > GuttaFlow 2 > RoekoSeal, as also shown by the trend of the A_2160_/A_2963_ and A_910_/A_1258_ absorbance ratios (Figure 2).

The spectrum of the white paste of GuttaFlow Bioseal differs from the spectra of the other white components for the presence of a broad shoulder at about 930 cm^−1^ assignable to the non-bridging oxygen, Si–O_NBO_ stretching mode [25] of the bioactive silicate glass (i.e., 45S5 [8]). If detected, the bands ascribable to vinyl groups in white components (i.e., C=C stretching at 1600 cm^−1^ and =CH stretching at 3055 cm^−^^1^) were observed only as weak spectral features, in agreement with Cai et al. [21]. These bands decreased in intensity along with the series: GuttaFlow Bioseal > GuttaFlow 2 > RoekoSeal.

Figure 3 shows the average IR spectrum of fresh (i.e., just mixed) GuttaFlow Bioseal; the spectra of its white and orange pastes are reported for comparison. Analogous figures for RoekoSeal and GuttaFlow 2 are reported in the (Appendix A).

As can be easily seen, the Si-H bands at 2160 and 910 cm^−1^ decreased in intensity upon mixing, as observed from the trend of the above-mentioned A_2160_/A_2963_ and A_910_/A_1258_ ratios (Figure 2). This trend agrees with the mechanism of the hydrosilylation reaction [20,21,26] discussed below (see Section 3). The band assigned to the bioactive glass was no longer observed in the spectrum of fresh GuttaFlow Bioseal due to the superposition of the 910 cm^−1^ band.

Figure 4 shows the average FT-Raman spectra of orange (A) and white (B) pastes of RoekoSeal, GuttaFlow 2, and GuttaFlow Bioseal. Band wavenumbers and assignments are summarized in Appendix A.

All the FT-Raman spectra are dominated by the bands of monoclinic zirconia [24,27]; the strongest bands of the organic phase were observed at 2967, 2908, and 2870 cm^−1^. In the spectra of the white components, the bands of PDMS were observed as weak spectral features at 1411, 1265, 861, 790, 755, 710, and 690 cm^−1^ [21,28]. Among these bands, the spectra of the orange pastes show only some spectral features due to the bad fluorescence background. In the spectra of the white pastes, the bands ascribable to vinyl groups (i.e., C=C stretching at 1600 cm^−1^ and =CH stretching at 3055 cm^−1^) were observed only as very weak spectral features, in agreement with Cai et al. [21]. The FT-Raman spectrum of the white component of GuttaFlow Bioseal differs from the spectra of the other white components for the presence of broad bands at about 1080 and 960–940 cm^−1^, assignable to the Si-O-Si stretching of SiO_4_ tetrahedra and PO_4_ stretching [29] of the bioactive silicate glass (i.e., 45S5 [12]).

The FT-Raman spectra of the orange pastes of GuttaFlow 2 and GuttaFlow Bioseal show a distinctive band at about 1670 cm^−1^, assignable to gutta-percha. This band was not detected in the spectra of the corresponding white pastes, suggesting that this phase is present only in the orange component. No 1670 cm^−1^ band was observed in the spectra of RoekoSeal, in agreement with the material composition declared by the manufacturer. The relative intensity of the band at 1670 cm^−1^ suggests that GuttaFlow Bioseal should contain a higher amount of gutta-percha than GuttaFlow 2. This result is confirmed by the trend of the FT-Raman spectra of the fresh sealers (Figure 5); the band at 1670 cm^−1^ was detected only in the spectrum of GuttaFlow Bioseal. It is not surprising that the spectra quality worsened with respect to those of the white pastes, and only some bands of the PDMS component were detected due to the contribution of the bad background of the orange component. The FT-Raman spectra of GuttaFlow 2 and GuttaFlow Bioseal (orange pastes and fresh samples, Figure 4A and Figure 5) show a weak band at 435 cm^−1^, assignable to zinc oxide [30]. This component was not revealed for RoekoSeal, in agreement with a previous EDX study [10].

The intensity ratio between the bands of PDMS and those of zirconia decreased along with the series: RoekoSeal ≈ GuttaFlow Bioseal > GuttaFlow 2, as also shown by the trend of the I_710_/I_638_ intensity ratio (Figure 6).

The micro-Raman spectra confirmed these trends (Appendix A). Both orange and white pastes contained monoclinic zirconia and PDMS in all the commercial sealers. The relative content of these two phases was investigated through the I_710_/I_638_ intensity ratio as in FT-Raman spectra (Appendix A). Gutta-percha was detected only in the spectra of the orange component of GuttaFlow 2 and GuttaFlow Bioseal, in agreement with the FT-Raman findings. It is interesting to note that the micro-Raman spectra showed bands at about 1020, 1000, 970, 590, 510, and 455 cm^−1^ that were not observed in the previously reported FT-Raman spectra (Figure 4 and Figure 5, see Appendix A for comparison, as an example). This behavior may be ascribed to the difference in the laser excitation wavelength (1064 nm in the FT-Raman spectra and 532 nm in the micro-Raman ones). Bands at similar wavenumber values were also observed in BioRoot RCS [16].

#### 2.1.2. Doped Samples

Appendix A, shows the IR spectrum of fresh GuttaFlow 2 + 20% BioRoot RCS; the spectra of fresh GuttaFlow 2 and BioRoot RCS are reported for comparison. Some spectral features ascribable to BioRoot RCS were detected between 850 and 1000 cm^−1^. Some bands assigned above to PDMS appeared to shift compared to fresh GuttaFlow 2 due to the contribution of the mineralizing agent. The A_910_/A_1258_ absorbance ratio was calculated for the commercial samples (Appendix A). Its value did not appear significantly different if compared with GuttaFlow 2 (*p* > 0.05), although it could be overestimated because of the contribution of BioRoot RCS to the 910 cm^−1^ band.

Appendix A, shows the IR spectrum of fresh GuttaFlow 2 + 20% DCPD; the spectra of fresh GuttaFlow 2 and DCPD are reported for comparison. Only weak spectral features appeared ascribable to DCPD in the spectrum of the doped sample. The A_910_/A_1258_ absorbance ratio appeared significantly higher (*p* < 0.05) than in undoped GuttaFlow 2 (Appendix A), although the contribution of the DCPD to the 910 cm^−1^ band cannot be excluded.

Appendix A shows the IR spectrum of fresh GuttaFlow 2 + 20% HA; the spectra of fresh GuttaFlow 2 and HA are reported for comparison. Several spectral features assignable to HA were detected in the spectrum of the doped sample; the A_910_/A_1258_ absorbance ratio did not appear significantly different (*p* > 0.05) if compared with undoped GuttaFlow 2 (Appendix A).

Micro-Raman and FT-Raman spectra of the fresh doped cements were not significantly different compared to those of fresh GuttaFlow 2, and the doping agents were not detected. The micro-Raman spectra of the fresh samples will be reported in the following section and were used as controls to detect the formation of a mineral deposit.

### 2.2. Bioactivity Tests

Figure 7 and Figure 8 show the average IR and micro-Raman spectra recorded on the surface of GuttaFlow Bioseal before (i.e., fresh) and after aging in HBSS for 28 days, respectively.

The IR spectrum of the aged sample showed distinct bands at 1455–1414 cm^−1^ (ν_3_ CO_3_^2−^ antisymmetric stretching), 1015 cm^−1^ (ν_3_ PO_4_^3–^ antisymmetric stretching), 868 cm^−1^ (ν_2_ CO_3_^2−^ out-of-plane bending), and 600–559 cm^−1^ (ν_4_ PO_4_^3−^ out-of-plane bending), all assignable to a B-type carbonated apatite [31,32]. Upon aging, the PDMS band observed in fresh cement at 790 cm^−1^ increased its relative intensity and shifted to 793 cm^−1^. The calculation of the fourth derivative IR spectra (see inset of Figure 7) allowed us to understand that this trend was ascribable to the appearance of a new component at 800 cm^−1^, assignable to a silica-rich layer (Si-O-Si symmetric stretching/bending [33,34]).

The micro-Raman spectra (Figure 8) confirmed the formation of a calcium phosphate deposit through the strengthening of the 971 cm^−1^ band in the spectrum recorded using a 3000 μm pinhole and even more with a 50 μm pinhole. The IR A_1010_/A_1258_ absorbance ratio and the Raman I_970_/I_638_ intensity ratio were chosen as calcium phosphate deposition spectroscopic markers; Figure 9 shows their values before and after aging for all materials under study. As can be easily seen, for GuttaFlow Bioseal, both ratios increased significantly (*p* < 0.05) upon aging.

Appendix A, show the average IR and micro-Raman spectra recorded on the surface of RoekoSeal before (i.e., fresh) and after aging in HBSS for 28 days, respectively. Regarding IR spectra, only a slight strengthening was detected near 560, 1000, and 1400 cm^−1^, where the strongest bands of calcium phosphates/apatites and carbonates fall. No significant changes in the shape and relative intensity of the PDMS bands were detected (Appendix A), suggesting that the polymer did not undergo any significant degradation under these conditions.

The micro-Raman spectra (Appendix A) did not disclose any strengthening in the 950–1000 cm^−1^ (phosphate) or 1050–1100 cm^−1^ (carbonate) ranges, and this behavior was also observed when using a pinhole of 50 μm, i.e., under experimental conditions more sensitive to the sample surface. As can be easily seen, for RoekoSeal, the IR A_1010_/A_1258_ absorbance ratio and the Raman I_970_/I_638_ intensity ratio (Figure 9) did not significantly increase (*p* > 0.05).

Appendix A, show the average IR and micro-Raman spectra recorded on the surface of GuttaFlow 2 before (i.e., fresh) and after aging in HBSS for 28 days, respectively. The IR spectra showed a behavior similar to RoekoSeal; on the contrary, the average micro-Raman spectrum recorded with a 3000 μm pinhole showed a strengthening of the band at 972 cm^−1^. To verify the hypothesis that this increase was effectively due to the formation of a deposit, the same sample areas were analyzed also using a 50 μm pinhole. The further increase in the intensity of the 972 cm^−1^ band under these experimental conditions confirmed the hypothesis. However, the above-mentioned IR and Raman ratios (Figure 9) increased, but not significantly (*p* > 0.05).

Appendix A show the average IR spectra recorded on the surface of the doped cements (i.e., GuttaFlow 2 + 20% BioRoot RCS, GuttaFlow 2 + 20% DCPD, GuttaFlow 2 + 20% HA, respectively) before (i.e., fresh) and after aging in HBSS for 28 days, respectively. The corresponding micro-Raman spectra are reported in Appendix A. The FT-Raman spectra did not reveal any deposit (spectra not shown).

The IR spectra of the aged-doped materials showed an increase in the intensity of the above-mentioned marker bands of B-type carbonated apatites; accordingly, the IR A_1010_/A_1258_ absorbance ratio (Figure 9) increased significantly (*p* < 0.05) for all the samples. Upon aging, the PDMS band observed in fresh cement at about 790 cm^−1^ increased its relative intensity and shifted to higher wavenumber values; this behavior was particularly evident for GuttaFlow 2 + 20% BioRoot RCS and GuttaFlow 2 + 20% HA.

The micro-Raman spectra confirmed the formation of a calcium phosphate deposit, as revealed by the strengthening of the 970 cm^−1^ band already discussed and the increase of the I_970_/I_638_ intensity ratio (Figure 9). However, the high standard deviation associated with its values made differences not significant (*p* > 0.05).

## 3. Discussion

Vibrational spectroscopic techniques appear to be suitable complementary tools to gain more insights into the composition of commercial sealers. IR spectroscopy allowed us to characterize the PDMS component and the reaction occurring upon mixing the orange and white pastes. As observable from Figure 1, the white paste was found to contain a PDMS “base material”, which was characterized by the presence of vinyl groups also according to the Raman spectra; the orange paste was found to contain a “coupling agent” characterized by the presence of Si-H bonds, well-identified through the IR bands at about 2160 and 910 cm^−1^. According to the literature [20,21,26], several PDMS crosslinked polymers may be produced by mixing vinyl end-capped oligomeric or polymeric dimethyl siloxane, an alkyl hydrogen siloxane as a crosslinking agent, and an organo-metallic platinum compound as a catalyst for the hydrosilylation reaction [20,21,26]. As can be seen from the IR spectra, in all the sealers, the above-mentioned Si-H bands decreased in intensity upon mixing. This trend agrees with the mechanism of the hydrosilylation reaction [20,21,26]: as reported in Figure 10, vinyl and silicon hydride groups react, forming Si-CH_2_-CH_2_-Si linkages between the PDMS chains, leading to crosslinking.

As can be seen from the IR spectra, the Si-H bands weakened but did not disappear, suggesting that a fraction of the coupling agent remained unreacted.

IR spectroscopy showed that the orange and white pastes contained different amounts of the reactive groups in the three sealers. In particular, it was observed that the bands assignable to both Si-H and vinyl groups decreased in intensity going along the series: GuttaFlow Bioseal > GuttaFlow 2 > RoekoSeal; this behavior appears particularly clear by analyzing the trend of the A_2160_/A_2963_ and A_910_/A_1258_ IR absorbance ratios (identified as markers of the Si-H content), which significantly (*p* < 0.05) decreased along the same series (Figure 2). Moreover, the data reported in Figure 2 show that the concentration of residual Si-H groups in the fresh samples was higher for GuttaFlow 2 than for GuttaFlow Bioseal; for RoekoSeal, no Si-H residual groups were detected upon curing. However, it must be observed that in this cement, their content was the lowest among the analyzed sealers.

These results allow us to explain the trend of the setting time, which was found to increase along the same series as above, i.e., GuttaFlow Bioseal < GuttaFlow 2 < RoekoSeal [10], suggesting that a lower concentration of reactive species slows down the setting process. It is well known that the initial concentration of the reactive species (i.e., Si-H and vinyl-containing species) influences the reaction rate, its completion, and the degree of crosslinking [35], which in turn affect stiffness, adhesion, and mechanical properties [36,37].

By comparing the values of the A_910_/A_1258_ IR absorbance ratios in the fresh samples (Appendix A), it can be observed that in the doped cements (i.e., added with BioRoot RCS, DCPD, HA), this marker had tendentially higher values (although not significantly, in most cases) than in undoped GuttaFlow 2. This behavior would suggest that fewer crosslinks are formed in the composites in the presence of the doping agents; evidently, the doping agents somehow interacted with the components of the sealer, slowing down the reaction between the reactive sites. Similar effects have been reported in the literature [13].

Both IR and FT-Raman spectroscopies were able to detect the presence of a bioactive glass-ceramic component in GuttaFlow Bioseal, according to the composition declared by the producer.

Raman spectroscopy proved suitable to gain useful information on the relative amounts of organic and inorganic components. This technique allowed us to also detect zirconia (which was found to be in its monoclinic polymorph) and gutta-percha. The former phase was present in both white and orange pastes, while the latter was identified only in the orange paste.

Based on the I_710_/I_638_ values obtained from the FT-Raman spectra (Figure 6), it can be affirmed that GuttaFlow 2 contained a significantly higher (*p* < 0.05) relative amount of zirconia than both RoekoSeal and GuttaFlow Bioseal, which did not show significantly different contents of this inorganic phase between each other (*p* > 0.05). Micro-Raman measurements showed an analogous trend (Appendix A), although the higher standard deviations associated with the I_710_/I_638_ ratios made differences between materials not significant (*p* > 0.05). The Raman results on relative zirconia contents agree with the previously reported radiopacity measurements [10]: GuttaFlow Bioseal and RoekoSeal had good (i.e., fulfilling ISO recommendations) and comparable radiopacity values (5.62 and 5.60 mm Al, respectively), significantly lower than GuttaFlow 2 (8.16 mm Al).

To gain more insights into the apatite-forming ability of the sealers, the commercial and doped cements were aged in HBSS for 28 days. The used aging conditions did not affect the structure of the cement matrix. This result did not appear unexpected since PDMS and gutta-percha are known to degrade upon aging under more severe conditions and for longer times [38,39,40,41].

The IR A_1010_/A_1258_ absorbance ratio and the Raman I_970_/I_638_ intensity ratios (Figure 9) proved to be suitable markers to compare the samples’ bioactivity: the bands at about 1010 cm^−1^ (IR) and 970 cm^−1^ (Raman) are both assignable to calcium phosphate vibrations. Both IR and micro-Raman vibrational techniques agreed on the fact that among the analyzed samples, GuttaFlow Bioseal showed the highest bioactivity (with the deposition of a B-type carbonated apatite), whilst this property was lacking for RoekoSeal and negligible for GuttaFlow 2.

Regarding the latter result, it may be observed that earlier studies have disclosed that PDMS-based biomaterials were undesirably characterized by a certain ability to calcify in in vitro and in vivo experiments, especially in the presence of fatty acids [42,43]. More recent investigations have reported that PDMS shows no bioactivity [8,14] unless it is mixed with bioceramics [8,9,14]. In confirmation, the introduction of PDMS in polyurethanes has been used as a good strategy to prevent their calcification [44,45]. The higher bioactivity observed for GuttaFlow 2 with respect to RoekoSeal may be explained by the different compositions of the sealers revealed by the vibrational measurements. As reported above, RoekoSeal was found to contain lower amounts of Si-H and vinyl-containing crosslinking reactive sites than GuttaFlow 2, suggesting a lower crosslinking degree in the former material. The polymer network present in GuttaFlow 2 probably allows a better three-dimensional arrangement of the groups favoring calcium adsorption/complexation. Moreover, it must be recalled that, in agreement with a previous study [10], GuttaFlow 2 contained zinc oxide while RoekoSeal did not. Zinc has been widely used in dentistry since it plays an important role in mineralization [46]; it improves the bioactivity of the dentin tissue [47], increases its resistance to an acidic environment [48], and inhibits collagen degradation [49]. The importance of the sealer matrix is indirectly confirmed by considering the calcium release data previously published for the commercial sealers under study [10]; it is interesting to note that after 28 days of aging, the commercial sealers released in water comparable amounts of Ca^2+^ ions (i.e., 22 ± 8 ppm for GuttaFlow Bioseal, 16 ± 6 ppm for GuttaFlow 2, and 11 ± 3 ppm for RoekoSeal [10]). Nevertheless, they displayed a noticeably different behavior in HBSS, suggesting that the ability of the matrix to interact with calcium ions plays a significant role in determining bioactivity.

The higher bioactivity displayed by GuttaFlow Bioseal agrees with its water sorption data, which were significantly higher than for both GuttaFlow 2 [10,11] and RoekoSeal [10]. This result may be explained in relation to the well-known water-mediated ion exchange occurring at the 45S5 bioglass surface when placed in physiological fluids [50,51] and suggests that the bioglass particles are exposed from the hydrophobic PDMS matrix so that they are able to hydrate. The appearance of the IR component at about 800 cm^−1^ in the spectra of the aged GuttaFlow disk (Figure 7) spectroscopically confirms that upon hydration [52], a SiO_2_-rich layer formed on the surface of the bioglass. The Si-OH groups act as nucleation sites for calcium and phosphate ions; mineralization early involved the formation of amorphous calcium phosphates, which transform into hydroxyapatite and then into hydroxycarbonate apatite, able to favor bone cells attachment and proliferation. In agreement with this mechanism, the IR spectra reported in Figure 7 showed the marker bands of a B-type carbonated apatite. The bands of PDMS were still detected, indicating that the deposit was not thick enough to mask the cement underneath.

An analogous mechanism should be hypothesized to occur also for the doped cements; the results discussed below confirm this hypothesis.

All of the doped cements exhibited apatite-forming ability, although to a lower extent than the most bioactive GuttaFlow Bioseal; as observed in Figure 9, they showed lower increases in the IR A_1010_/A_1258_ absorbance ratio (significant, *p* < 0.05 for all) and Raman I_970_/I_638_ intensity ratio (not significant, *p* > 0.05 for all). Among the doped cements, the most bioactive appeared that added with 20% BioRoot RCS; upon aging, the % increase in the IR A_1010_/A_1258_ absorbance ratio was the highest, i.e., 112% (Figure 9A). The bioactivity of the doped cements did not appear unexpected: CaSi, DCPD, and HA, which appeared well distributed within the cement disks, evidently acted as apatite nucleating sites in the poorly bioactive GuttaFlow 2 matrix.

Regarding the sealer doped with BioRoot RCS, the apatite deposit formed on it was thinner and more amorphous than that revealed on pure BioRoot RCS under the same conditions [16]; in the micro-Raman spectra recorded on GuttaFlow 2 + 20% BioRoot RCS after 28 days of aging in HBSS (Appendix A), the apatite band at about 965 cm^−1^ had a lower relative intensity and was broader than in pure BioRoot RCS treated under the same conditions [16]. Moreover, it may be observed that the % increase in the IR A_1010_/A_1258_ absorbance ratio was lower than for GuttaFlow Bioseal (Figure 9A), although BioRoot RCS was found to release a significantly higher calcium amount (i.e., nearly 60 times higher) than GuttaFlow Bioseal upon aging for 28 days in water [10,16]. Evidently, the PDMS matrix slowed down the CaSi hydration and, thus, calcium release, which in turn influenced bioactivity. It is well known that CaSi-based cements form apatite when immersed in physiological fluids thanks to their calcium-releasing ability upon hydration. The alite and belite components hydrate, producing calcium hydroxide, whose release provokes a pH increase, and a calcium silicate hydrate (CSH) phase, a porous structure containing Si-OH groups, which at alkaline pH are prevalently deprotonated as SiO^−^ groups [53]. Both Si-OH and SiO^−^ may act as heterogeneous nucleation sites for apatite deposition, according to a previously proposed model [54,55]. On the other hand, silicon-containing nanocomposites have been incorporated in several medical implants and bone grafts thanks to the relevant biological role played by this trace element in mineralization processes [56,57], being associated with an increased bone density and decreased bone loss [58] due to stimulation of osteoblast activity and osteoclasts inhibition [59], and activation of genes that enhance bone proliferation [56].

Interestingly, the shift and strengthening of the PDMS IR band at 790 cm^−1^ upon aging were observed not only in the bioglass-containing (i.e., silica-forming) GuttaFlow Bioseal. Analogous behaviors were observed for GuttaFlow 2 + 20% BioRoot RCS and GuttaFlow 2 + 20% HA, and to a significantly lower extent for GuttaFlow 2 + 20% DCPD (Appendix A), in agreement with the lower bioactivity of the latter sample, as disclosed from the lower % increase in the IR A_1010_/A_1258_ absorbance ratio (Figure 9A). This trend would suggest that, upon mineralization, the PDMS matrix rearranged; it must be recalled that Si-O-Si symmetric stretching modes fall in the spectral range near 800 cm^−1^. On the other hand, in PDMS-HA composites, some authors have identified the band at 800 cm^−1^ as a marker of the interaction between the two components [13]. Therefore, this band may be interpreted accordingly also in the spectra of our aged samples as a sign of the role played by the matrix in calcium phosphate enucleation.

Bioactivity tests, aimed to investigate the apatite-forming abilities of biomaterials, have been proposed by Kokubo et al. and other research groups [60,61,62]. It should be evidenced that the ability to form apatite plays a critical role in helping osteoblasts to produce a new bone matrix [63,64]. As clinical implications, endodontic sealers may be extruded over the apex and may be in contact with the periapical bone defects that require new bone formation. Using apatite-forming/bioactive sealers is innovative in endodontic therapy since apatite formation also contributes to dentin remineralization, according to other studies [65,66].

## 4. Materials and Methods

### 4.1. Materials

Tested commercial sealers were RoekoSeal (Coltène/Whaledent GmbH, Langenau, Germany), GuttaFlow 2 (Coltène/Whaledent AG, Altstätten, Switzerland), and GuttaFlow Bioseal (Coltène/Whaledent AG, Altstätten, Switzerland). All the sealers are two-components systems made of an orange paste and a white paste. According to the manufacturer, RoekoSeal contains polydimethylsiloxane (PDMS), silicone oil, paraffin-based oil, zirconium dioxide, and a platinum catalyst. GuttaFlow 2 contains gutta-percha powder, PDMS, a platinum catalyst, zirconium dioxide, microsilver (preservative), and coloring. GuttaFlow Bioseal contains gutta-percha powder, PDMS, a platinum catalyst, zirconium dioxide, silver (preservative), coloring, and a bioactive glass-ceramic. Therefore, it appears that the latter sealer has a composition similar to GuttaFlow 2 but contains, in addition, a bioactive glass-ceramic.

In an attempt to improve material bioactivity, GuttaFlow 2 was added with 20% by weight (*w*/*w*) dicalcium phosphate dihydrate (DCPD) powder (Sigma-Aldrich, Steinheim, Germany) or 20% *w*/*w* hydroxyapatite (HA) powder (Sigma-Aldrich, Steinheim, Germany) or 20% *w*/*w* BioRoot RCS (Septodont, Saint-Maur-des-Fossés, France). The latter sealer is a two-component cement constituted by a powder containing tricalcium silicate, zirconium oxide, povidone, and an aqueous solution of calcium chloride and polycarboxylate. It was prepared according to the manufacturer’s instructions.

Material disks were prepared using PVC molds (8.0 ± 0.1 mm diameter × 1.6 ± 0.1 mm thickness).

### 4.2. Bioactivity Tests

Demolded material disks were immediately immersed vertically in 20 mL of HBSS (Hank’s Balanced Salt Solution, Lonza Walkersville, Inc., Walkersville, MD, USA) used as simulated body fluid and stored at 37 °C for 28 days. The medium was renewed weekly [67,68]. Simple material disks were preferred to cylindrical molds open on one side to reduce the interferences provided by the mold material. However, a model using root dentin samples inserted into a cylinder open to one side only confirmed the abundant production of apatite in the extruded apical sealer [69].

### 4.3. Raman and IR Analyses

The surfaces of fresh samples and after aging in HBSS for 28 days were analyzed by IR and Raman vibrational spectroscopies. Orange and white components of RoekoSeal, GuttaFlow 2, and GuttaFlow Bioseal were analyzed for comparison, as well as DCPD, HA, and BioRoot RCS.

IR spectra were recorded in triplicate on a Bruker Alpha Fourier Transform FTIR spectrometer, equipped with a Platinum Attenuated Total Reflectance (ATR) single reflection diamond module (penetration depth 2 μm) and a Deuterated Lanthanum α-Alanine-doped TriGlycine Sulfate (DLaTGS) detector; the spectral resolution was 4 cm^−1^.

FT-Raman spectra were recorded in triplicate using a Bruker MultiRam FT-Raman spectrometer equipped with a cooled Ge-diode detector. The excitation source was an Nd^3+^-YAG laser (1064 nm) in the backscattering (180°) configuration. The focused laser beam diameter was about 100 μm, the spectral resolution was 4 cm^−1^, and the laser power at the sample was about 80 mW.

Micro-Raman spectra were obtained using an NRS-2000C Jasco spectrometer with a microscope of 100× magnification and a pinhole with an aperture diameter of 3000 μm. Five spectra at least were recorded on each sample and averaged. All the spectra were recorded in backscattering conditions with 5 cm^−1^ spectral resolution using the 532 nm green diode-pumped solid-state laser (RgBLase LLC, Fremont, CA, USA) with a power of about 20 mW. A 160 K cooled digital charge-coupled device (Spec-10: 100B, Roper Scientific Inc., Sarasota, FL, USA) was used as a detector. A confocal pinhole with an aperture diameter of 50 μm was placed in the optical circuit to obtain signals from a limited in-depth region. Unless specified, the spectra shown are recorded with a 3000 μm pinhole.

The two Raman techniques were used as complementary tools to gain information on the composition of the fresh samples and the mineral deposit formed upon aging in HBSS. The micro-Raman technique appeared more suitable to detect the formation of a mineral deposit on the surface of the samples after aging in HBSS, while the FT-Raman technique proved more useful to gain information on the relative amount of the different phases in the commercial samples.

### 4.4. Statistical Analysis

The statistical analysis (on Raman and IR data) was performed with R statistical software (version 3.5.3; GNU GPL license). The data analysis was conducted through the one-way analysis of variance (ANOVA). The means comparison was carried out by Tukey’s HSD test (set at *p* < 0.05).

## 5. Conclusions

Vibrational IR and Raman spectroscopies were successfully used to characterize (at a molecular level) three endodontic sealers based on polydimethylsiloxane, i.e., RoekoSeal, GuttaFlow 2, and GuttaFlow Bioseal. The differences in composition and the crosslinking degree were related to relevant properties in the endodontic practice, such as the setting time and radiopacity, as well as the apatite-forming ability.

Aging tests in HBSS showed that GuttaFlow Bioseal was the most bioactive sample, with the spectroscopically revealed deposition of a B-type carbonated apatite.

The PDMS matrix was found to play a relevant role in the bioactivity of GuttaFlow 2; evidently, the crosslinking degree was favorable for the occurrence of Ca^2+^ adsorption/complexation and the formation of a thin calcium phosphate layer. In the less crosslinked RoekoSeal, such processes did not occur.

The doped cements showed bioactivity higher than GuttaFlow 2 base material; this result indicates that the particles of the mineralizing agents (i.e., BioRoot RCS, DCPD, HA) are spontaneously exposed on the cement surface due to significant differences between their hydrophilicity and the hydrophobicity of the PMDS matrix. If PDMS totally covers the re-mineralizing particles, their dissolution reactions providing the desired biomineralization are hindered. On the other hand, the hydrophobicity of the PDMS matrix was found to slow down the process of apatite deposition.

## Figures and Tables

**Figure 1 molecules-27-05750-f001:**
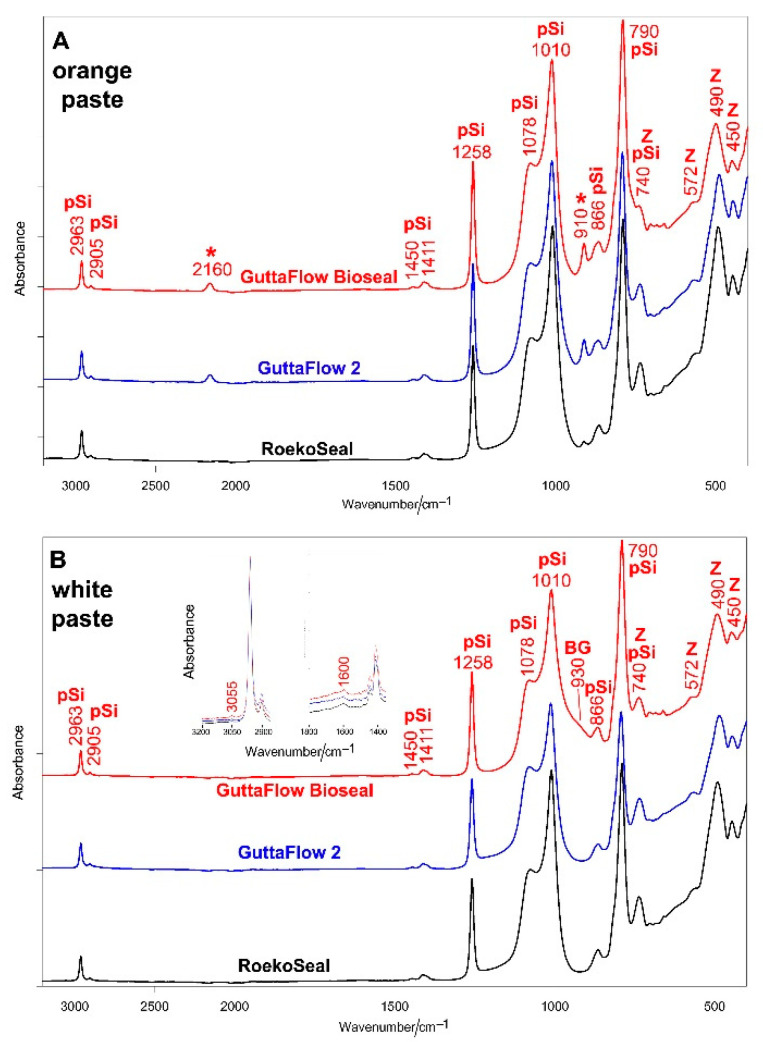
The average IR spectra of orange (**A**) and white (**B**) pastes of RoekoSeal, GuttaFlow 2, and GuttaFlow Bioseal. The spectra are normalized to the absorbance of the 2963 cm^−1^ band. The bands assignable to polydimethylsiloxane (pSi), monoclinic zirconia (Z), and bioactive glass-ceramic (BG) are indicated together with those specifically assigned to Si-H bonds (*). The insets show the spectral ranges where the modes ascribable to vinyl groups are reported to fall (i.e., C=C stretching at about 1600 cm^−1^ and =CH stretching at about 3055 cm^−1^). More detailed assignments are summarized in Appendix A.

**Figure 2 molecules-27-05750-f002:**
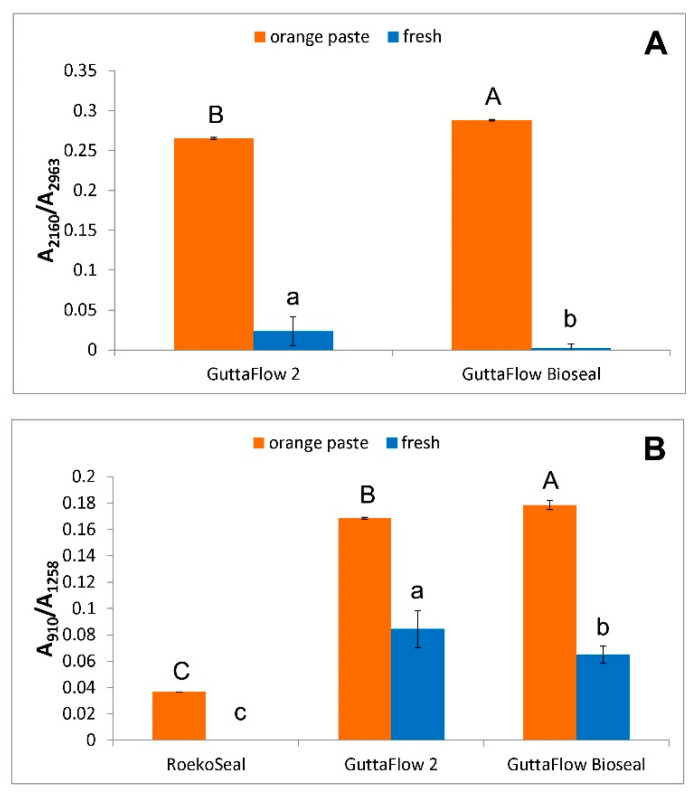
A_2160_/A_2963_ (**A**) and A_910_/A_1258_ (**B**) absorbance ratios (average ± standard deviation) as calculated from the IR spectra of the orange paste and fresh (i.e., just mixed) samples of the commercial sealers under study. The different capital letters in each histogram represent statistically significant differences (*p* < 0.05) between orange pastes and small letters between the fresh samples.

**Figure 3 molecules-27-05750-f003:**
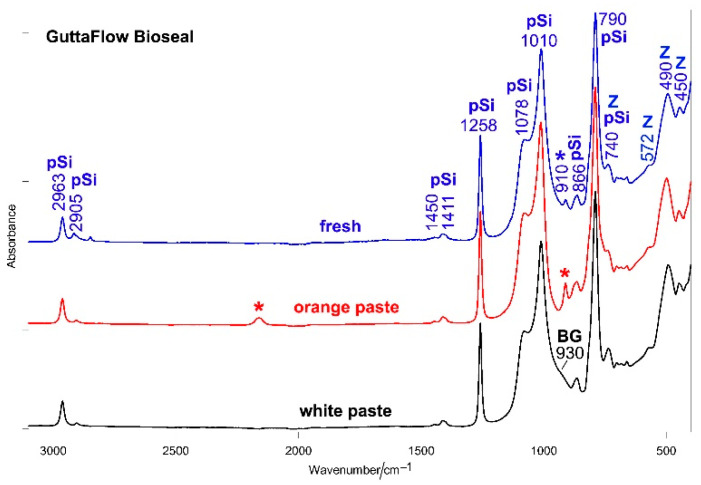
The average IR spectrum of fresh (i.e., just mixed) GuttaFlow Bioseal; the spectra of its white and orange pastes are reported for comparison. The spectra are normalized to the absorbance of the 2963 cm^−1^ band. The bands assignable to polydimethylsiloxane (pSi), monoclinic zirconia (Z), and bioactive glass-ceramic (BG) are indicated together with those specifically assigned to Si-H bonds (*). More detailed assignments are summarized in Appendix A.

**Figure 4 molecules-27-05750-f004:**
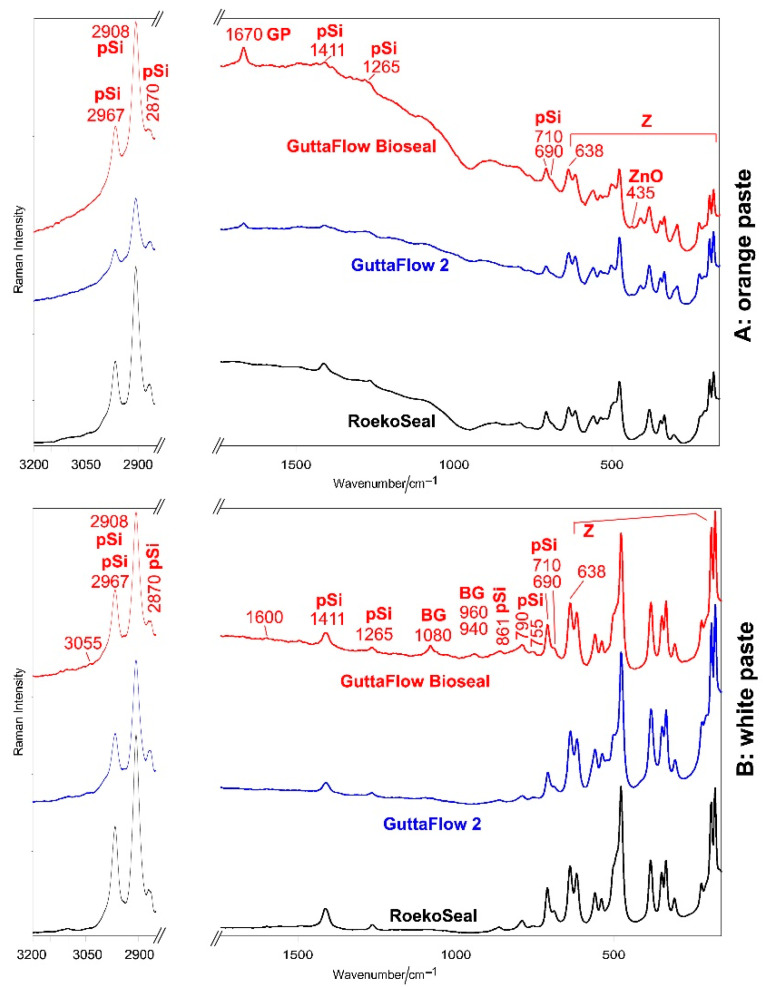
Average FT-Raman spectra of orange (**A**) and white (**B**) pastes of RoekoSeal, GuttaFlow 2, and GuttaFlow Bioseal. The spectra are normalized to the intensity of the 638 cm^−1^ band. The bands assignable to monoclinic zirconia (Z), polydimethylsiloxane (pSi), bioactive glass-ceramic (BG), zinc oxide (ZnO), and gutta-percha (GP) are indicated. More detailed assignments are summarized in Appendix A.

**Figure 5 molecules-27-05750-f005:**
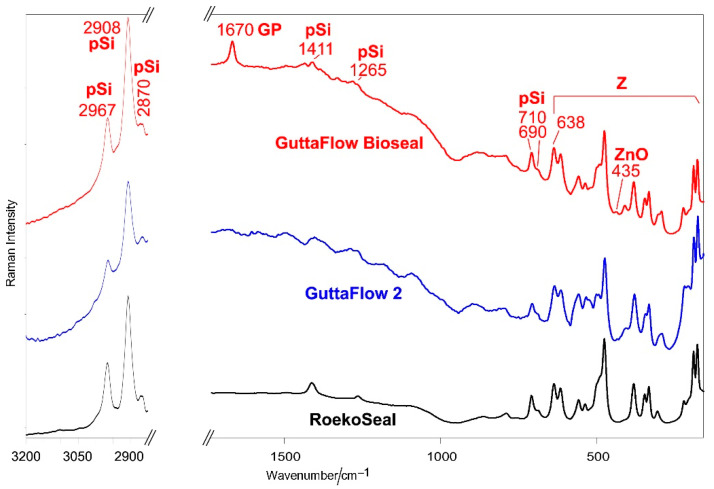
Average FT-Raman spectra of fresh RoekoSeal, GuttaFlow 2, and GuttaFlow Bioseal. The spectra are normalized to the intensity of the 638 cm^−1^ band. The bands assignable to monoclinic zirconia (Z), polydimethylsiloxane (pSi), zinc oxide (ZnO), and gutta-percha (GP) are indicated. More detailed assignments are summarized in Appendix A.

**Figure 6 molecules-27-05750-f006:**
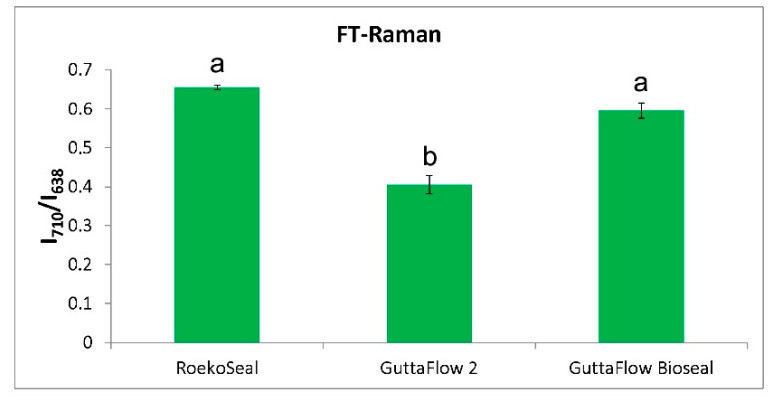
I_710_/I_638_ intensity ratio (average ± standard deviation) as calculated from the FT-Raman spectra of fresh commercial sealers under study. Different letters represent statistically significant differences (*p* < 0.05) between values.

**Figure 7 molecules-27-05750-f007:**
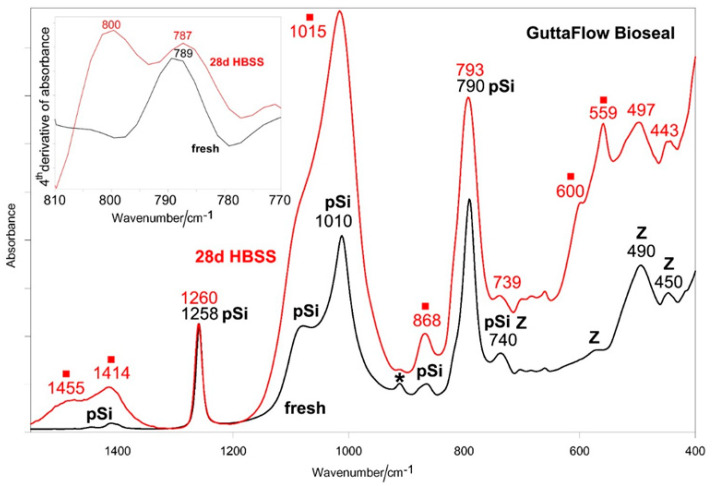
Average IR spectra recorded on the surface of GuttaFlow Bioseal before (i.e., fresh) and after aging in HBSS for 28 days. The spectra are normalized to the absorbance of the 1258 cm^−1^ band. The bands assignable to polydimethylsiloxane (pSi) and monoclinic zirconia (Z) are indicated together with those specifically assigned to unreacted Si-H bonds (*) and B-type carbonated apatite (■). The inset shows the fourth derivative IR spectra in the 810–770 cm^−1^ range. More detailed assignments are reported in the text and summarized in Appendix A.

**Figure 8 molecules-27-05750-f008:**
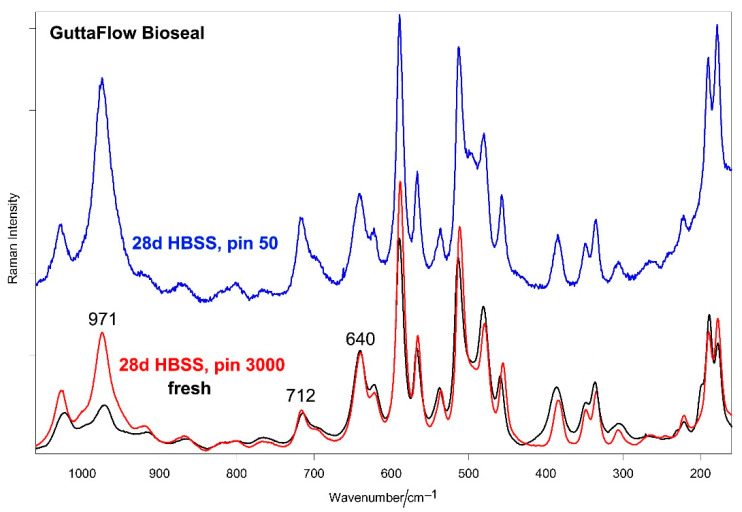
Average micro-Raman spectra recorded on the surface of GuttaFlow Bioseal before (i.e., fresh) and after aging in HBSS for 28 days. The spectra were normalized to the intensity of the 638 cm^−1^ band. The spectra of the aged sample were recorded using pinholes of 3000 μm (pin 3000) and 50 μm (pin 50). The bands assignable to monoclinic zirconia (Z) and polydimethylsiloxane (pSi) are indicated.

**Figure 9 molecules-27-05750-f009:**
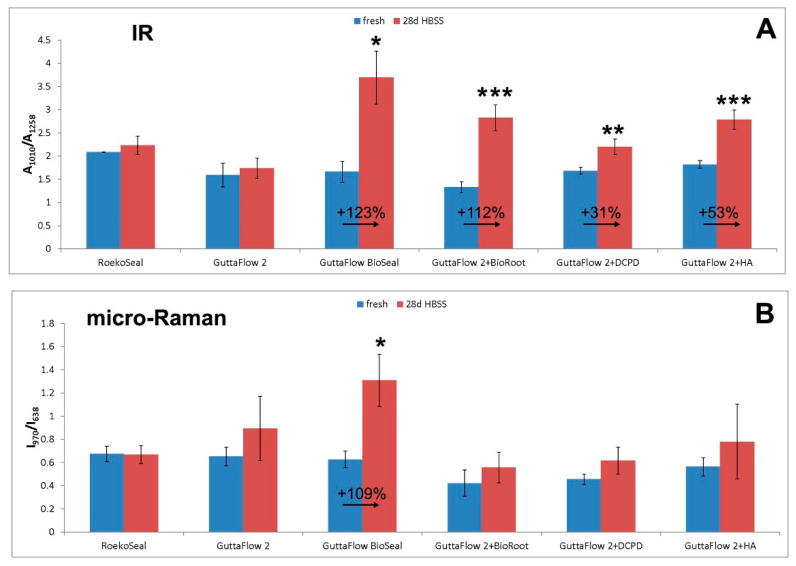
IR A_1010_/A_1258_ (**A**) absorbance ratio, and Raman I_970_/I_638_ (**B**) intensity ratio (average ± standard deviation) as calculated from the IR and micro-Raman spectra of the materials under study before (i.e., fresh samples) and after aging for 28 days in HBSS. Asterisks indicate significant differences (*: *p* < 0.01; **: *p* < 0.001; ***: *p* < 0.0001) between fresh and aged samples within the same material in a Tukey’s HSD test.

**Figure 10 molecules-27-05750-f010:**
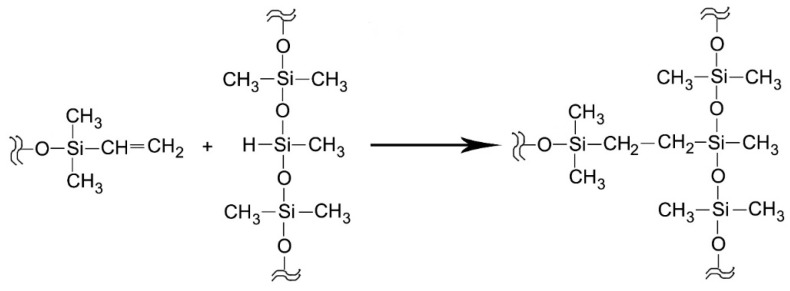
Mechanism of the hydrosilylation reaction.

## Data Availability

From the authors.

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
