# Peer review of "The Influence of the Matrix on the Apatite-Forming Ability of Calcium Containing Polydimethylsiloxane-Based Cements for Endodontics"

_molecules, 2022, doi:10.3390/molecules27185750_

Round 1
Reviewer 1 Report
Taddei et al. analyzed the characteristics of endodontic sealers at a molecular level using vibrational IR and Raman spectroscopy. The analyses were detailed and sophisticated, and the contents were scientifically stated.
However, it is difficult to explain how this research can contribute to clinical practice.
Major comments
1) Please state how the results of this research can contribute to clinical dentistry in Discussion and Conclusion (clinical significances). Especially, I cannot understand why the authors analyzed the orange and white paste separately. Clinicians are not interested in physical properties prior to mixing as they use the mix. The setting time and radiopacity are provided by a company.
2) Methods- 4.2. Bioactivity tests
Although the analyses were detailed and complex, the methods used in this study were very simple. Material disks were prepared using a mold. And the disk was immersed vertically in HBSS. This bioactivity test does not reflect actual clinical practice. The sealer should be evaluated, for example, on a cylindrical mold that are open on one side only, like previous investigations. Results would be overestimated because a bare sealer block would leach out a lot of the active ingredients.
Author Response
1) Please state how the results of this research can contribute to clinical dentistry in Discussion and Conclusion (clinical significances). Especially, I cannot understand why the authors analyzed the orange and white paste separately. Clinicians are not interested in physical properties prior to mixing as they use the mix. The setting time and radiopacity are provided by a company.
Response: With regards to the contribution of this research to clinical dentistry, a paragraph has been added at the end of the Discussion section, with pertinent references:
“Bioactivity tests, aimed to investigate the apatite forming ability of biomaterials, have been proposed by Kokubo et al. and other research groups [60-62]. It should be evidenced that the ability to form apatite plays a critical role in helping osteoblasts to produce a new bone matrix [63,64]. As clinical implications, endodontic sealers may be extruded over the apex and may be in contact with the periapical bone defects that require new bone formation. Using apatite forming/bioactive sealers is innovative in endodontic therapy since apatite formation also contributes to dentine remineralization, according to other studies [65,66].”
With regards to the analyses of orange and white pastes, we added a sentence at the beginning of the Results section:
“Although clinicians use the sealers as the mix of the two pastes, we decided to analyze them separately to gain information on the relative amounts of the reactive sites present in the different cements and thus on the crosslinking degree.”
2) Methods- 4.2. Bioactivity tests
Although the analyses were detailed and complex, the methods used in this study were very simple. Material disks were prepared using a mold. And the disk was immersed vertically in HBSS. This bioactivity test does not reflect actual clinical practice. The sealer should be evaluated, for example, on a cylindrical mold that are open on one side only, like previous investigations. Results would be overestimated because a bare sealer block would leach out a lot of the active ingredients.
Response: We did not use any cylindrical mold. Simple material disks were used in many other studies to reduce the interferences provided by the mold material. However, a model with the use of root dentine samples, a cylinder open to one side only (as proposed by the referee) was previously reported [Prati et al. 2014, added as reference 69] and confirmed the abundant production of apatite in the extruded apical sealer.
A sentence in the experimental part has been added accordingly:
“Simple material disks were preferred to cylindrical molds open on one side to reduce the interferences provided by the mold material. However, a model using root dentine samples inserted into a cylinder open to one side only confirmed the abundant production of apatite in the extruded apical sealer [69].”
Reviewer 2 Report
Comments:
1. The present abstract highlights the results. Normally an abstract should include the above along with stating briefly the purpose of the study undertaken and meaningful conclusions based on the obtained results. I would expect a brief, yet concise quantitative description in the abstract.
2. L16 - endodontic sealer (GuttaFlow Bioseal) based on polydimethylsiloxane (PDMS) and containing a calcium bioglass as doping agent - What was the chemistry involved between all these molecules/materials? Any analytical proof?
3. L37 – rephrase the sentence - Root canal therapy is aimed at treating the infected root canal as well as at preventing..
4. L39 - Several endodontic sealers have been designed, what are those, enlist with examples and supporting references.
5. Figure 3 and onwards – I what authors can explain about the main peaks between 2000 and 500 wavenumbers.? Major peak numbers should be assigned with representative functional groups.
6. Results – Apart from IR, Data are not statistically shown and thus any results could not be interpreted well. The figures and tables not showing the statistical differences. Author should perform the statistical analysis by Tukey or Duncan test and indicate the significance in the superscript (a, b, c or in asterisk, *, **, **) of each value.
Author Response
- The present abstract highlights the results. Normally an abstract should include the above along with stating briefly the purpose of the study undertaken and meaningful conclusions based on the obtained results. I would expect a brief, yet concise quantitative description in the abstract.
Response: The purpose of the study and methods were stated more clearly; the results section was made more quantitative.
- L16 - endodontic sealer (GuttaFlow Bioseal) based on polydimethylsiloxane (PDMS) and containing a calcium bioglass as doping agent - What was the chemistry involved between all these molecules/materials? Any analytical proof?
Response: The cement composition was given according to manufacturer’s declaration.
- L37 – rephrase the sentence - Root canal therapy is aimed at treating the infected root canal as well as at preventing..
Response: The sentence was rephrased.
- L39 - Several endodontic sealers have been designed, what are those, enlist with examples and supporting references.
Answer: The different classes of endodontic sealers and the most representative examples are now reported in the introduction. References have been included:
“Zinc oxide-eugenol-based sealers (such as Pulp Canal Sealer and Argoseal) were some of the most clinically used materials with cold obturation techniques. These sealers demonstrated some drawbacks, including cytotoxicity when extruded over the apex [3] and inhibition of the polymerization phases of adjacent methacrylate-based materials [4]. Resin-based sealers are divided into methacrylate-based or epoxy resin-based. Methacrylate-based sealers (such as Endorez) are no longer used as demonstrated critical in vivo degradation, with loss of the endodontic seal and long-term failures [5]. Differently, epoxy resin-based sealers (such as AH Plus) are widely used in warm obturation techniques. As drawbacks, these sealers are highly hydrophobic and require the absence of moisture into the root canal [3]. Glass ionomer-based sealers (such as Ketac Endo) demonstrated biocompatibility and the ability to bond to dentin, but they are no longer used due to detrimental leakage and disintegration [3]. Calcium hydroxide-based sealers (such as Seal Apex or Apexit) revealed high biocompatibility, but are seldomly used nowadays due to their solubility [6].”
- Figure 3 and onwards – I what authors can explain about the main peaks between 2000 and 500 wavenumbers.? Major peak numbers should be assigned with representative functional groups.
Response: Tables reporting bands assignments have been added as Supplementary Material (Tables S1 and S2).
- Results –Apart from IR, Data are not statistically shown and thus any results could not be interpreted well. The figures and tables not showing the statistical differences. Author should perform the statistical analysis by Tukey or Duncan test and indicate the significance in the superscript (a, b, c or in asterisk, *, **, **) of each value.
Response: The authors would like to remark that the statistical analysis of all the spectroscopic ratios discussed in the main text was already present in the original paper. Anyway, following reviewer’s suggestion, we repeated the statistics using Tukey HSD’s test, setting P < 0.05. Accordingly, the text was modified at lines 521-527. The spectroscopical data subjected to statistical analysis are shown in Figures 2, 6, 9 (in the main text) and Figures S6 and S9 (in the Supplementary materials), with the proper indication of significance. It’s worth to mention that the spectroscopic ratios shown in Figure 9 were additionally analysed setting P at different values (0.0001 < P < 0.01).
Round 2
Reviewer 1 Report
The authors revised this manuscript according to the reviewer's suggestion.
Reviewer 2 Report
The revised version reads well. Authors have addressed all the comments raised in the last review. This manuscript can now be accepted for publication.